# Integrated omics approach to unveil antifungal bacterial polyynes as acetyl-CoA acetyltransferase inhibitors

Ching-Chih Lin[1,2,6], Sin Yong Hoo [1,2,6], Li-Ting Ma[1,2,6], Chih Lin[1], Kai-Fa Huang [3], Ying-Ning Ho[4], Chi-Hui Sun[1], Han-Jung Lee[1], Pi-Yu Chen[1], Lin-Jie Shu[1], Bo-Wei Wang[1,2,5], Wei-Chen Hsu[1,2], Tzu-Ping Ko [3] & Yu-Liang Yang [1,2✉]

Bacterial polyynes are highly active natural products with a broad spectrum of antimicrobial activities. However, their detailed mechanism of action remains unclear. By integrating comparative genomics, transcriptomics, functional genetics, and metabolomics analysis, we identified a unique polyyne resistance gene, *masL* (encoding acetyl-CoA acetyltransferase), in the biosynthesis gene cluster of antifungal polyynes (massilin A **1**, massilin B **2**, collimonin C **3**, and collimonin D **4**) of *Massilia* sp. YMA4. Crystallographic analysis indicated that bacterial polyynes serve as covalent inhibitors of acetyl-CoA acetyltransferase. Moreover, we confirmed that the bacterial polyynes disrupted cell membrane integrity and inhibited the cell viability of *Candida albicans* by targeting ERG10, the homolog of MasL. Thus, this study demonstrated that acetyl-CoA acetyltransferase is a potential target for developing antifungal agents.

[1] Agricultural Biotechnology Research Center, Academia Sinica, Nankang Dist., Taipei 115, Taiwan. [2] Biotechnology Center in Southern Taiwan, Academia Sinica, Guiren Dist., Tainan 711, Taiwan. [3] Institute of Biological Chemistry, Academia Sinica, Nankang Dist., Taipei 115, Taiwan. [4] Institute of Marine Biology and Center of Excellence for the Oceans, National Taiwan Ocean University, Jhongjheng Dist., Keelung 202, Taiwan. [5] Department of Marine Biotechnology and Resources, National Sun Yat-sen University, Gushan Dist., Kaohsiung 804, Taiwan. [6] These authors contributed equally: Ching-Chih Lin, Sin Yong Hoo, Li-Ting Ma. ✉email: ylyang@gate.sinica.edu.tw

Polyynes or polyacetylenes contain a conformationally rigid rod-like architecture and an electron-rich consecutive acetylene moiety. Hundreds of polyynes have been discovered, out of which compounds have mainly been isolated from terrestrial plants and marine organisms[1,2]. In contrast to polyynes from plant sources, bacterial polyynes contain a distinct terminal alkyne with conjugated systems, which causes bacterial polyynes to be more unstable. This instability has discouraged surveys of bacterial polyynes using a bioactivity-guided isolation approach. Nevertheless, more than ten bacterial polyynes have been recorded in a few species. Notably, most polyynes have been reported to have a broad spectrum of antimicrobial effects. For instance, cepacins, isolated from *Pseudomonas cepacian* SC 11783 (taxonomically reclassified as a *Burkholderia diffusa*), was reported to have antibacterial activity against the majority Gram-negative bacteria and *Staphylococci* spp., and anti-oomycetal activity against *Pythium ultimum*[1–4]. Caryoynencin, isolated from *Pseudomonas caryophylli* (taxonomically reclassified as *Burkholderia caryophylli*), was reported to have antibacterial activity against *Escherichia coli*, *Klebsiella pneumoniae*, *Staphylococci aureus*, and *Bacillus subtilis*[5]. In addition, collimonins isolated from *Collimonas fungivorans* Ter331[6,7] and Sch 31828 isolated from *Microbispora* sp. SCC1438[8] were reported to have antifungal activity against *Aspergillus niger* and *Candida* spp., respectively.

Other polyynes, such as ergoynes isolated *Gynuella sunshinyii* and fischerellins from *Fischerella muscicola*, were found to have potential cytotoxic/ antibacterial and allelopathic ability[9–11]. Recently, protegenins (including protegencin) produced by *Pseudomonas protegens* were reported to have antifungal, anti-oomycete, and anti-algae activities[12–15]. Despite the apparent bioactivity of these compounds, the active target and mechanism(s) remain unclear.

Here, we used a multi-omic approach and genetic engineering to identify the bioactive polyynes of *Massilia* sp. YMA4, characterized their biosynthesis gene cluster (BGC) named massilin (*mas*) BGC, and delineated the antifungal mechanism of bacterial polyynes. Furthermore, heterologous co-expression of *mas* BGC further illustrated the conserved genes critical to the electron-rich consecutive alkyne moiety and an additional gene responsible for structure derivatives modification of the polyynes. Furthermore, a comparative genomic analysis uncovered the self-resistance gene and identified bacterial polyynes as antifungal agents that target the first enzyme of ergosterol biosynthesis, acetyl-CoA acetyltransferase. Finally, crystallographic analysis unveiled the detailed binding model of polyynes to the acetyl-CoA acetyltransferase.

## Results and discussion

**Transcriptomics analysis reveals polyynes as antifungal agents and their encoding BGC in *Massilia* sp. YMA4.** An antagonism assay of *Massilia* sp. YMA4 against *C. albicans* revealed a distinct phenotype showing that the antifungal metabolites were produced in potato dextrose agar (PDA) medium but not in yeast malt agar (YMA) medium (Supplementary Fig. 1). In addition, the antifungal metabolites were unstable in the extract and laborious to purify for bioassays using the classic bioactivity-guided isolation approach, which includes purification with click chemistry, exclusion of oxygen and light, or careful optimization[7,12,16]. Therefore, to reveal the antifungal metabolites and their BGCs, we used a combined transcriptomics and metabolomics approach to explore the compounds produced in the two different media (PDA and YMA). First, the BGCs in the genome of *Massilia* sp. YMA4 were characterized via DeepBGC[17] with default settings and integrated with the criteria: DeepBGC score > 0.7 to characterize 13 BGCs in *Massilia* sp. YMA4 (Supplementary Data 1). Then, the differentially expressed genes (192 upregulated and 226 downregulated with *P*-value < 0.05 and |fold change| > 2) from transcriptomics analysis of PDA vs. YMA cultured cells were mapped into the 13 characterized BGCs. Among them, only one BGC was upregulated in PDA compared to YMA (Supplementary Data 1), and we named this gene cluster massilin (*mas*) BGC with 12 transcribed genes (*masA* to *masL*) (Fig. 1a).

To characterize the metabolites produced from *mas* BGC, we constructed mutant strains through insertion mutation at the *masH* (encoding fatty acid desaturase) gene locus in *Massilia* sp. YMA4 (Supplementary Fig. 2). The result of the antagonism assay illustrated that YMA4::*masH* lost antifungal activity against *C. albicans* (Supplementary Fig. 3), which demonstrated that *mas* BGC was crucial for the antifungal activity of *Massilia* sp. YMA4. Furthermore, a study of polyyne BGC (*pgn* BGC) in *P. protegens* had determined that the mutation of homolog (*pgnH*) of *masH* and *masI* in *pgn* BGC can effectively disrupt the production of protegencin (also known as protegenin A)[12]. We then conducted mass spectrometry-based untargeted metabolomics to analyze the differential features in wild type and biosynthesis-deficient mutant strain YMA4::*masH*. Via target isolation of differential features, we purified four polyynes (collimonin C/D **1**, **2**, and massilin A/B **3**, **4**) from ethyl acetate extract of *Massilia* sp. YMA4 (Fig. 1b, c). Their structures were elucidated using high-

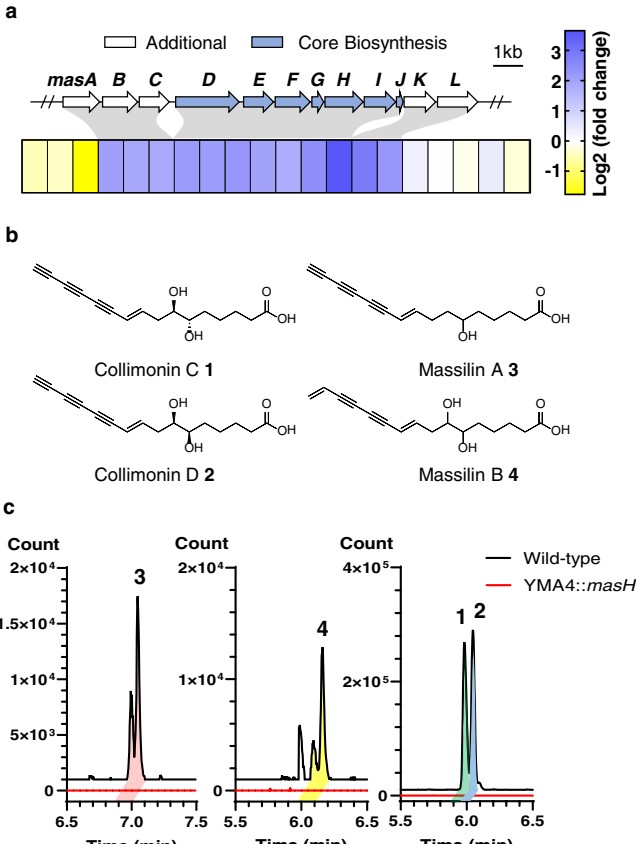

**Fig. 1 Polyynes and polyyne biosynthesis gene cluster of *Massilia* sp. YMA4. a** Gene expression profile around the *mas* BGC under polyyne production (PDA) versus non-production (YMA) medium. The expressions of additional and core biosynthesis genes of *mas* BGC are framed in the gray area. **b** Structure of polyynes collimonin C/D **1**, **2** and massilin A/B **3**, **4**. **c** Extraction ion chromatography (EIC) of collimonin C/D **1**, **2** (*m/z* 273.1132), massilin A **3** (*m/z* 257.1183), and massilin B **4** (*m/z* 275.1289) in *Massilia* sp. YMA4 wild type (black) and the biosynthesis null mutant strains (YMA4::*masH*) with a 10 ppm mass window.

resolution mass spectrometry (Supplementary Figs. 4 and 5) and nuclear magnetic resonance (Supplementary Tables 1–4 and Supplementary Figs. 6–21; the detailed isolation procedure and structure elucidation are described in Supplementary Methods). Among the four polyynes identified from *Massilia* sp. YMA4, collimonin C **1** and collimonin D **2** were initially isolated from *C. fungivorans* Ter331[7]. A new compound with an ene-triyne moiety was named massilin A **3** and identified as a racemate at C(6)OH of an unsaturated hexadecanoic acid. Another new ene-diyne-ene compound, massilin B **4**, was presumed to be the precursor of collimonin C **1** or collimonin D **2**. Notably, massilin B **4** was more chemically stable than other polyynes with a terminal alkyne.

In antifungal activity assay (Supplementary Fig. 22), polyynes with a terminal alkyne showed potent inhibition of *C. albicans* with minimum inhibitory concentrations (MIC): 69.73 µM (collimonin C **1**), 35.24 µM (collimonin D **2**), and 2.40 µM (massilin A **3**). However, massilin B **4** with a terminal alkene moiety had no antifungal activity with MIC > 500 µM. These results suggest that the terminal alkyne is essential for polyyne antifungal activity and stability.

**Phylogenetic analysis of polyyne BGCs and *mas* heterologous co-expression revealed the core components for polyyne biosynthesis.** The in-silico mining and comparative studies of polyynes/ terminal alkyne containing BGCs indicated that *jam*ABC homologs (fatty acyl-AMP ligase (FAAL), fatty acid desaturase (FAD), and acyl carrier protein (ACP)) in jamaicamides (*jam*) BGC would be the minimal module for polyynes biosynthesis[18,19]. In addition, the gene cluster K in *C. fungivorans* Ter331 (collimonin BGC, which is abbreviated to *col* BGC) recruited not only FAAL, FAD, and ACP but also hydrolase/ thioesterase (H/TE) and rubredoxin (Rd)[6,7]. Therefore, we further explored the architecture of 56 bacterial polyynes BGCs with *mas* BGC as a reference to confirm the conserved composition of polyyne biosynthetic associated enzymes. It contains the FAAL–2x FAD–ACP–FAD–H/TE–Rd (Supplementary Fig. 23).

The essential proteins, including FAAL, FAD, and H/TE in the conserved region of polyyne BGCs, have been confirmed in the biosynthesis of caryoynencin in *B. caryophylli*[16], cepacins in *Burkholderia ambifaria*[4], and protegenins in *P. protegens*[12,13]. To further verify the function of other core biosynthetic genes, we use plasmid-derived insertion mutagenesis to gain serial mutants of *mas* BGC from *masD* (encoded FAAL) to *masL* (Supplementary Figs. 2 and 3). YMA4::*masD*, YMA4::*masE*, YMA4::*masF*, YMA4::*masH*, and YMA4::*masI* exhibited completely disrupted polyyne biosynthesis. YMA4::*masJ* exhibited significantly decreased production of polyynes. However, YMA4::*masK* and YMA4::*masL* did not affect the biosynthesis process.

To complement the mutagenesis results, we further established an *E. coli* heterologous co-expression system of *mas* genes to determine whether the genes participate in polyyne biosynthesis. The engineered *E. coli mas*– (heterologous co-expression with *masD* to *masJ*) produced a relatively nonpolar polyyne (Fig. 2a). This new ene-triyne polyyne was named massilin C **5** (Supplementary Table 5 and Supplementary Figs. 24–28). The results demonstrated that *masD* to *masJ* are the core genes in the biosynthesis of the ene-triyne moiety. On the other hand, the expression of *masB* in *Massilia* sp. YMA4 was upregulated when cultured in PDA medium (Supplementary Data 1), like the core genes *masD* to *masJ*. Therefore, we proposed *masB*, a dioxygenase gene, as a candidate for the hydroxylation of polyynes since cytochrome P450s or dioxygenases can execute similar reactions[20,21]. The *masB* incorporated strain, *mas* + , produced collimonin C/D **1**, **2** (Fig. 2b and Supplementary Fig. 24), which

suggested massilin C **5** is a biosynthetic intermediate and further hydroxylated by MasB to generate polyynes **1**–**4**. Furthermore, the absence of *masJ* significantly decreased the accumulation of polyynes in *mas* expressed *E. coli* (*mas*–' and *mas* + ', Fig. 2), indicating the crucial role of rubredoxin in polyyne biosynthesis. As an electron transporter, rubredoxin can shuttle reducing molecules from NAD(P)H to membrane-bound alkane hydroxylases[22], which is likely crucial to activation/regeneration of acetylenase/desaturase for the poly-dehydrogenation process in polyyne biosynthesis.

**MasL serves as a polyyne direct target with a protective function.** "Antibiotic resistome" refers to the notion that not only antibiotic resistance genes present in resistant pathogens but also those in the antibiotic producers often harbor resistance genes for self-protection[23,24]. Sometimes the resistance genes are located within the BGC of antibiotics[25]. To reveal the target protein of the bacterial polyynes of interest, we adopted the concept of the antibiotic resistome. We conducted phylogenetic analysis of bacterial polyyne BGCs and classified the BGCs into two chemotaxonomic groups: a palmitate-derived (C16) and a stearate-derived (C18) group (Fig. 3a and Supplementary Fig. 29). In the palmitate-derived group BGCs, cepacin (*ccn*) BGC branched out before the most recent common ancestor of *mas*/*col* BGCs.

Interestingly, the gene encoding the major facilitator superfamily (MFS) transporter, implicated in multidrug resistance and transports of small molecules and xenobiotics[26], is preserved in *col* BGC but lost in *mas* BGC (Fig. 3b, Supplementary Table 6, and Supplementary Data 2). This implies that *Massilia* sp. YMA4 has an alternative protection mechanism for self-protection from polyynes. Notably, a conspicuous gene *masL*, an acetyl-CoA acetyltransferase gene, remains in *mas* BGC and ancestor *ccn* BGC but is absent in *col* BGC. We, therefore, proposed that *masL* is a potential self-resistance gene of polyynes.

To evaluate the protective effect of *masL*, we first heterologously expressed *masL* in polyyne-sensitive *C. albicans*. The expression of *masL* rescued fungal cell viability from polyyne inhibition under minimum inhibitory concentration (Fig. 4 and Supplementary Table 7). Furthermore, an in vitro acetyl-CoA acetyltransferase (MasL) inhibition assay showed that polyynes (collimonin C/D **1**, **2**, and massilin A **3**) inhibited the MasL enzyme activity (Table 1 and Supplementary Fig. 30). Since the drug target identification could be achieved by gene overexpression for gain-of-function of resistance[27], we suggest *masL* serves as a self-resistance gene in the *mas* BGC, and the acetyl-CoA acetyltransferase it encodes is the target of collimonin C/D **1**, **2** and massilin A **3**. This targeting manner might explain why the growth of *mas* expressed *E. coli* is unaffected when producing polyynes without *masL* because *E. coli* has an acetyl-CoA acetyltransferase-independent route, the methylerythritol phosphate (also called deoxyxylulose 5-phosphate) pathway, to produce isoprenoid[28,29]. Moreover, *ccn* BGC divided into *mas* BGC and *col* BGC independently, and each contains a different self-protection mechanism, which implies a deletion event after horizontal gene transfer[30].

**Polyynes are covalent inhibitors of acetyl-CoA acetyltransferase.** To investigate how bacterial polyynes target acetyl-CoA acetyltransferase, we solved the crystal structures of MasL in its apo and two holo (bound collimonin C **1** and collimonin D **2**) forms at 1.78 Å, 1.66 Å, and 1.40 Å resolution, respectively (Table 2). The asymmetric unit (space group *P*1 for apo MasL and MasL-collimonin D complex; *P*2₁ for MasL-collimonin C complex) of both structures contains a tetramer of the protein (Fig. 5, Table 2 and Supplementary Fig. 31), as observed in

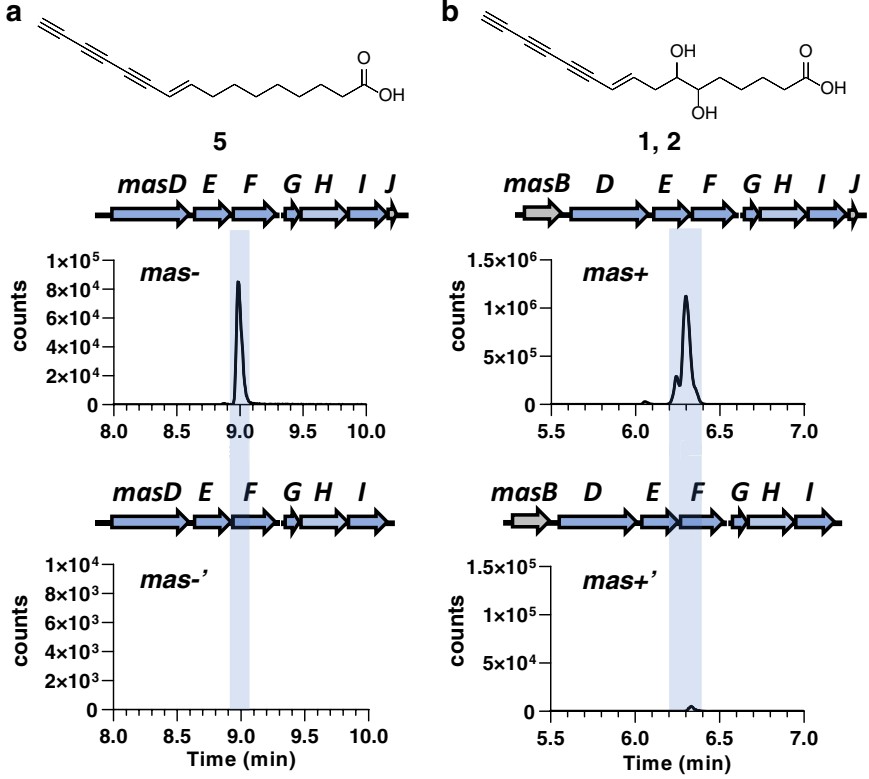

**Fig. 2 Heterologous co-expression of *mas* genes in *E. coli*. a** The EIC (*m/z* 241.1234) of engineered *E. coli* strain, which carried *masD*, *masE*, *masF*, *masG*, *Bv4687* (homolog of *masH* from *Burkholderia vietnamiensis* LMG 10929) and *masI* when co-expressed with (*mas*–, top) or without (*mas*–', bottom) *masJ*. **b** The EIC (*m/z* 273.1132) of *E. coli* strain *mas*– and *mas*–' harboring additional dioxygenase (*masB*) were renamed as *mas* + and *mas* + '. The structures of products are illustrated on top of each chromatogram and were verified by LC-HRMS/MS analysis (Supplementary Fig. 24).

solution. The monomer of MasL shares the general fold architecture reported in the type II biosynthetic thiolase family[31]. MasL consists of three domains: an N-terminal α/β domain (N-domain, residues 1–121 and 251–271), a loop domain (L-domain, residues 122–250), and a C-terminal α/β domain (C-domain, residues 272–394) (Supplementary Fig. 32). The N- and C-domains form a typical five-layered fold (α-β-α-β-α) as observed in the structures of the other type II biosynthetic thiolases including *Zoogloea ramigera* PhaA[31], *Clostridium acetobutylicum* CEA_G2880[32], and *Aspergillus fumigatus* ERG10A[33]. The L-domain displays an α/β fold with a tetramerization loop associated with the C-domain.

Many high-resolution atomic structural models of the substrate-binding complex revealed the Claisen condensation reaction and binding model within the reaction pocket of acetyl-CoA acetyltransferases. In apo and holo forms of MasL, the substrate-binding pocket was located on the enzyme's surface facing the opposite dimer of the tetrameric assembly. The pocket was a tunnel shape of ~10 Å depth with ~6–8 Å diameter for the linear pantothenic moiety of coenzyme A (CoA) extending through the reactive center. The reactive center in MasL contained reactive cysteine residues Cys90 and nucleophilic activation residues His350 and Cys380. In the MasL-collimonin C complex, the conjugated polyyne tail extended into the MasL substrate binding site and formed a covalent bond between the terminal carbon (C16) of the alkyne moiety and the reactive cysteine sulfhydryl moiety of Cys90 via nucleophilic addition (Supplementary Figs. 33 and 34). Thus, the terminal alkyne was essential for polyyne activity through covalent modification on the reactive cysteine sulfhydryl group of acetyl-CoA acetyltransferases.

Although we did not obtain a good X-ray diffraction result surrounding the reactive Cys90 in the MasL-collimonin D complex, the nucleophilic addition mechanism of polyynes to the acetyl-CoA acetyltransferase was confirmed by mass spectrometry analysis (Supplementary Fig. 35 and Supplementary Data 3). The observation of collimonin C/D ($+C_{16}H_{18}O_4$)–derived adducts of enzyme reactive cysteine residue provided convincing evidence of protein irreversible S-alkylation via nucleophilic addition to the conjugated terminal alkyne. Furthermore, the observation of massilin A ($+C_{16}H_{18}O_3$)–derived adducts suggest that massilin A **3** would also covalently inhibit acetyl-CoA acetyltransferase activity. In the further comparative analysis of the polar interactions in the activation pocket of MasL in MasL-collimonin C and MasL-collimonin D complex, the C(7)OH moiety of collimonin C/D **1, 2**, His158 of MasL, and a water molecule formed a strong polar interaction network, including a direct hydrogen bond (3.00–3.16 Å) and a water-mediated hydrogen bond between C(7) OH and His158 (Fig. 5). Furthermore, although there was no significant induced-fit within the pocket, the collimonin C/D **1, 2** caused the Arg135 on the tetramerization loop to swap and form a hydrogen bond interaction and salt bridge, respectively, across the two subunits within the binding site (Fig. 5). These polar interacting residues for the collimonin C/D **1, 2** (inhibitor) binding were similar to CoA (product) in other thiolase models[31]. Therefore, the acetyl-CoA acetyltransferase inhibitions by collimonin C/D **1, 2** are competitive binding models against substrate acetyl-CoA.

**The stereochemistry of hydroxyl groups on polyynes affects their binding affinity toward acetyl-CoA acetyltransferase.** The C6 chiral center of collimonin C/D **1, 2** is critical to forming different polar interactions. The C(6)OH of collimonin C **1** had

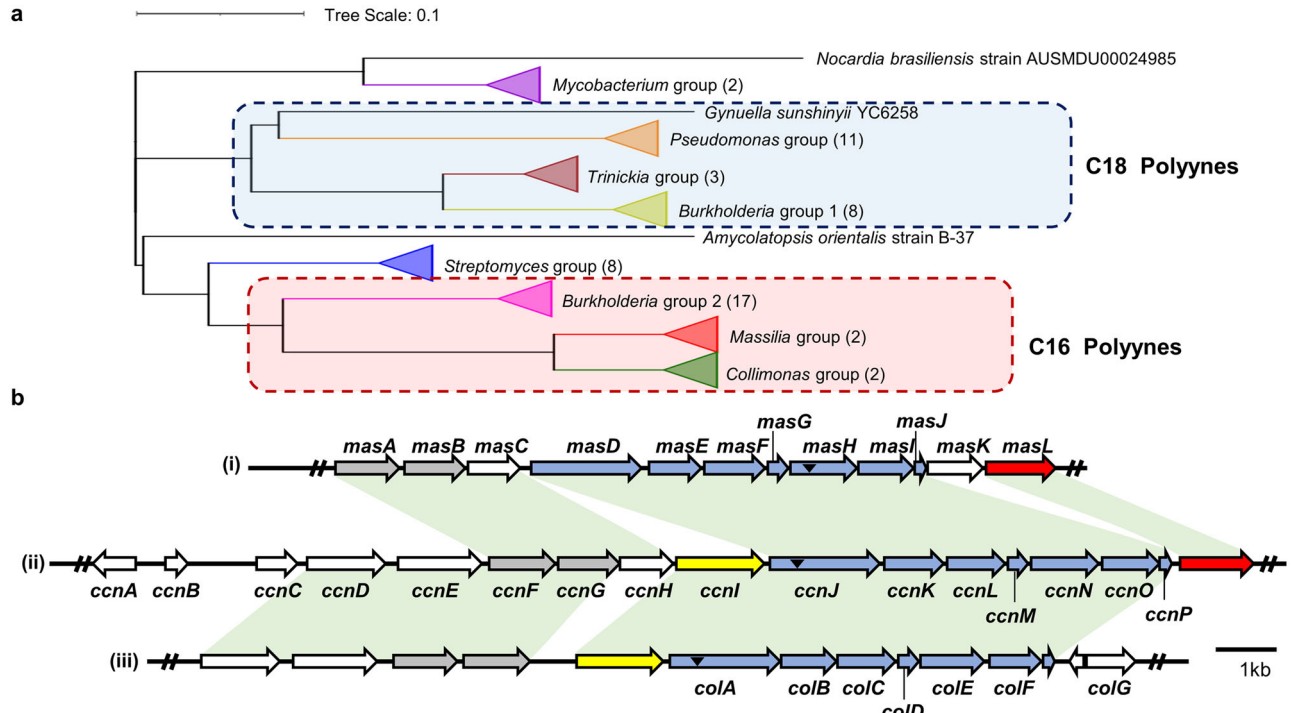

**Fig. 3 Comparative analysis of polyyne biosynthetic gene clusters and the structures of the bacterial polyynes. a** Phylogenetical analysis of polyyne biosynthetic gene clusters (BGCs) in bacteria (Supplementary Figs. 23 and 29). Species in the red box have been reported to produce palmitate-derived polyynes (C16), and species in the blue box have been reported to produce stearate-derived polyynes (C18). The number of BGCs in each group is shown in parentheses. **b** Comparison of the polyyne BGC architectures of *mas* BGC in *Massilia* sp. YMA4 (i), *ccn* BGC in *B. ambifaria* BCC0191(ii)[4], and *col* BGC in *C. fungivorans* Ter331(iii)[7]. Genes conserved in polyyne BGC across the phylogenetic tree are shown in blue, and those conserved in the C16 polyyne group are shown in gray. The potential protective genes in the BGC are shown in red for acetyl-CoA acetyltransferase and in yellow for MFS transporter. The corresponding homologs (over 40% identity) in BGCs between two species are shown as the light green area. Black triangles indicate the mutation sites in previous research and this study.

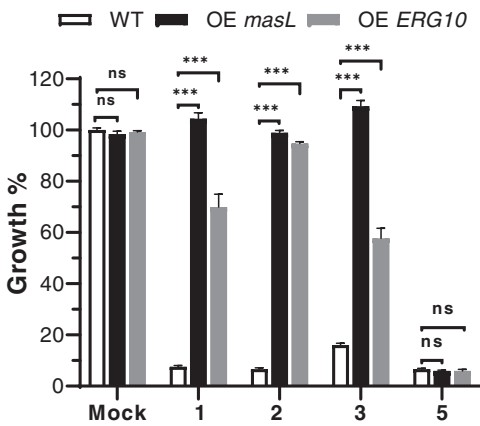

**Fig. 4 C. albicans was rescued by heterologous overexpression of masL (OE masL) and ERG10 (OE ERG10) from the minimum inhibitory concentration of collimonin C/D 1, 2, massilin A 3, and massilin C 5.** The cell viabilities were normalized to the mock treatment. The standard deviation was calculated based on three replicates and the two-tailed Student's t-test for statistical analysis. ***P-value < 0.000001. ns: not significant. Each experiment comprised three biological replicates (N = 3). All data values were listed in Supplementary Table 7.

**Table 1 Inhibition kinetic of *Massilia* sp. YMA4 MasL by polyynes.**

| Polyynes* | MasL | | |
|---|---|---|---|
| | $K_I$ (μM)[†] | $k_{inact}$ (min$^{-1}$)[†] | $k_{inact}/K_I$ (μM$^{-1}$ min$^{-1}$) |
| Collimonin C **1** | 297.10 | $9.798 \times 10^{-2}$ | $3.30 \times 10^{-4}$ |
| Collimonin D **2** | 42.84 | $5.208 \times 10^{-2}$ | $1.22 \times 10^{-3}$ |
| Massilin A **3** | 132.10 | $3.449 \times 10^{-2}$ | $2.61 \times 10^{-4}$ |
| Massilin C **5** | 228.00 | $5.477 \times 10^{-2}$ | $2.40 \times 10^{-4}$ |

*Each experiment contained three biological replicates (N = 3).
[†]Full data was illustrated in Supplementary Fig. 30.

more flexibility concerning the spatial direction (with a dihedral angle to C(7)OH from 109° to 170°) and built a sophisticated polar interaction network with the amide of Pro249 in the pantetheine loop, Arg135, and multiple water molecules (Fig. 5 and Supplementary Fig. 36). In contrast, the C(6)OH of collimonin D **2** did not form a polar interaction network at the contrasting face and showed fixed spatial direction (dihedral angle to C(7)OH from 32.9° to 42.6°) (Supplementary Fig. 36). The polar interaction network derived from C(6)OH of collimonin C **1** further affected the distal carboxyl group binding model. Furthermore, the reduced binding strength from the salt bridge (in MasL-collimonin D complex) to the hydrogen bond (in MasL-collimonin C complex) corresponded to their inhibition kinetic, that collimonin D **2** has a lower $K_I$ (42.84 μM) than collimonin C **1** (297.10 μM) (Table 1). The above information suggests that the stereochemistry of the hydroxyl group on polyynes is vital for initial non-covalent complex affinity. This salt bridge/hydrogen bond formation will dominate the MasL binding affinity in dihydro-type polyynes **1** and **2**.

**Table 2 X-ray data collection and refinement statistics.**

| | MasL | MasL-collimonin C complex | MasL-collimonin D complex |
|---|---|---|---|
| PDB code | 7EI3 | 7EI4 | 7FEA |
| Data collection | | | |
| Space group | *P1* | *P2₁* | *P1* |
| Cell dimensions | | | |
| *a, b, c* (Å) | 71.786, 76.995, 98.082 | 59.139, 115.077, 125.789 | 66.943, 77.741, 98.117 |
| α, β, γ (°) | 79.77, 79.00, 62.98 | 90.00, 91.08, 90.00 | 70.76, 81.78, 64.99 |
| Resolution (Å)* | 30.0–1.78 (1.84–1.78) | 30.0–1.66 (1.72–1.66) | 30.0–1.40 (1.45–1.40) |
| *R*merge | 0.045 (0.233) | 0.082 (0.560) | 0.056 (0.553) |
| *I* / σ*I* | 23.8 (5.0) | 17.7 (2.0) | 16.3 (2.0) |
| Completeness (%)* | 92.3 (85.2) | 99.4 (94.4) | 94.8 (86.0) |
| Redundancy* | 3.8 (3.9) | 5.6 (4.8) | 2.9 (2.5) |
| Refinement | | | |
| Resolution (Å) | 29.9–1.78 | 29.2–1.66 | 27.16–1.40 |
| No. reflections | 160,857 | 180,941 | 292,742 |
| *R*work/*R*free | 0.128/0.180 | 0.114/0.162 | 0.146/0.163 |
| No. atoms | | | |
| Protein | 11,522 | 11,544 | 11,808 |
| Ligand/ion | – | 80 | 80 |
| Water | 1,663 | 1,720 | 1,671 |
| B-factors | | | |
| Protein | 22.3 | 18.3 | 19.8 |
| Ligand/ion | – | 44.0 | 53.3 |
| Water | 34.9 | 34.1 | 32.6 |
| R.m.s. deviations | | | |
| Bond lengths (Å) | 0.008 | 0.009 | 0.015 |
| Bond angles (°) | 1.455 | 1.449 | 1.530 |

*Highest-resolution shell is shown in parentheses.*

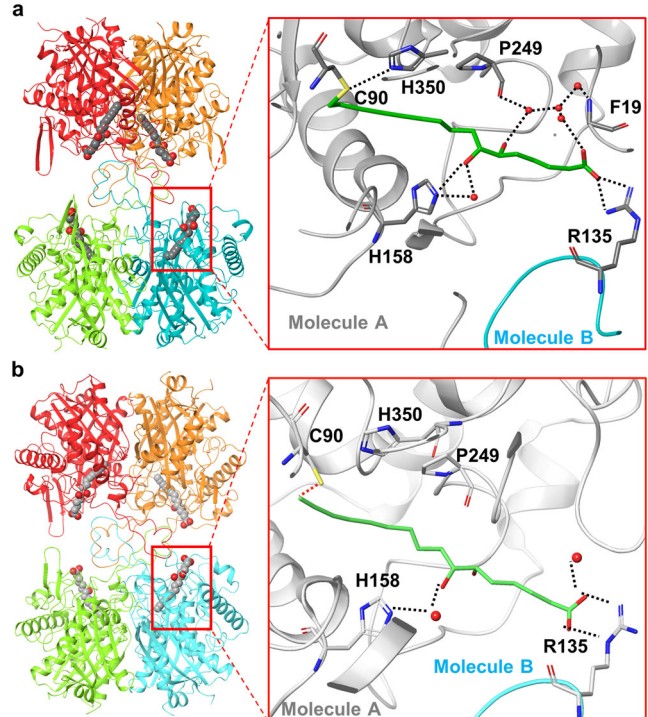

**Fig. 5 X-ray structures and polar interaction within the binding site of MasL-collimonin C and MasL-collimonin D complex.** Overall views of the crystal structures of MasL-collimonin C complex (**a**) and MasL-collimonin D (**b**). Four subunits per asymmetric cell are shown in red, orange, light green, and cyan, respectively. Collimonin C/D **1**, **2** is shown as grey space-filling balls. In the magnified view, the residues involved in collimonin C/D **1**, **2** interactions are shown as sticks with sequence identities indicated in the main chain molecule shown in gray. The R135 residue of another molecule is shown in cyan. The dotted lines indicate the hydrogen bonds and salt bridges involved in collimonin C/D **1**, **2** interactions within the binding pocket.

***Candida albicans* ERG10 is the antifungal target of polyynes**. As the bacterial polyynes target acetyl-CoA acetyltransferase MasL, we further evaluated the inhibition effect of bacterial polyynes on ERG10, an acetyl-CoA acetyltransferase homolog in *C. albicans*. ERG10 is the first enzyme of the ergosterol biosynthesis pathway and is crucial for fungal cell membrane formation. The enzyme activity of ERG10 was inhibited by polyynes **1–3** (Supplementary Fig. 37), and correspondingly the overexpression of ERG10 in *C. albicans* also rescued the cell viability from their inhibition (Fig. 4). Although we did not obtain the X-ray crystallography for the polyyne-ERG10 complex, the mass spectrometry analysis showed that polyynes **1–3** target the reactive Cys90 of ERG10 (Supplementary Fig. 38). A previous study reported that the homologous gene in *Aspergillus fumigatus*, *erg10A* is essential for survival and reduced expression leads to increased susceptibility to oxidative and cell wall-perturbing agents[33]. In the TEM analysis, we also observed that bacterial polyynes disrupted the cell membrane integrity in *C. albicans* (Supplementary Fig. 39). This result corresponded to the earlier finding that protegencin (or protegenin A) permeated algal cells and caused cell lysis[14]. Notably, massilin C **5**, the analog of protegencin (or protegenin A), possessed antifungal activity against *C. albicans* (MIC 12.22 μM, Supplementary Fig. 22) and a comparable ERG10 inhibitory potency to polyynes **1–3** (Supplementary Fig. 37). However, neither the heterologous expression of MasL nor ERG10 in *C. albicans* was able to recover the cell viability under massilin C **5** treatment (Fig. 4). Together these results suggested that the polyynes without a hydroxy moiety would possess multiple antifungal targets.

## Conclusion

The practical application of the antibiotic resistome concept with *in-silico* genome mining and comparative genetics analysis facilitated drug target identification. Our integrated approaches revealed that bacterial polyynes inhibit fungal growth by disrupting ergosterol biosynthesis. The crystallography analysis revealed that bacterial polyynes inhibit acetyl-CoA acetyltransferase via irreversible S-alkylation of the reactive residue with the functional terminal alkyne. In addition, the hydroxyl moiety stereochemistry affects the inhibitor affinity. We characterized the core biosynthesis module and modification genes for polyynes through genetic engineering. These findings will aid in the understanding of the structure-activity relationships of the polyynes and their production via microbial fermentation.

## Methods

**Bioinformatics analysis**. The biosynthesis gene clusters (BGCs) in the genome of *Massilia* sp. YMA4 (Genbank accession number **GCA_003293715.1**) were characterized via the command-line program DeepBGC[17] with default settings and integrated with DeepBGC score > 0.7 (Supplementary Data 2). Then, the *mas* BGC of *Massilia* sp. YMA4 was used to discover the homologous gene clusters in bacteria species using MultiGeneBlast[34] (Supplementary Data 2). The database was built using a bacterial sequences database (BCT, 2020 December 01) and whole-genome sequences of polyyne-reported bacterial species from NCBI. A total of 56 bacteria with polyyne BGC (Cumulative Blast bit score > 1500) were found. The homologous protein sequences of each bacterial polyyne BGC were respectively concatenated (a total of five amino sequences, starting from *masD* homolog to *masI* homolog). The concatenated protein sequences were used for alignment (MUSCLE) and the distance (UPGMA, bootstrap 5000 times) between 56 bacteria with

*Massilia* sp. YMA4 was identified for phylogenetic tree construction. The analysis was completed by using MEGA 10 with default parameters[35]. iTOL was used to present the results of phylogenetic analysis[36].

The RNA sequencing and transcriptome analysis were performed using Illumina MiSeq system (Illumina, USA) and CLC genomics workbench (version 11, CLC bio, Denmark), and the 92.89–98.11% reads were successfully mapped to the *Massilia* sp. YMA4 genome (GenBank assembly accession: GCA_003293715.1) for RPKM quantification (see Supplementary Methods and Supplementary Data 1 for details). The differential gene expression analysis was analyzed through CLC software with default pipeline and settings. Identification of differentially expressed genes (DEGs) with the expression (|Fold change (FC)| ≥ 2 with *P*-value < 0.05) was based on RPKM and analyzed using the Empirical analysis method[37]. Two biologic replicates of each condition were analyzed.

**Construction of *mas* co-expression platform in *Escherichia coli*.** Individual *mas* genes were cloned to the pET-28b(+) (Merck, Germany) using restriction enzyme digestion and ligation (see Supplementary Methods and Supplementary Tables 8–10 for details). To obtain functional *masH* in the heterologous *E. coli* system, we recruited native and codon-optimized *masH* and the homolog (*Bv*4687) from the palmitate-derived (C16) species *B. vietnamiensis* LMG 10929 (see Supplementary Fig. 23, 29 and 40) into modular co-expression. The backbone of the constructed modules, including pET-22a(+), pACYCduet, and pRSFduet, were purchased from Merck (Germany). However, polyyne accumulations were observed only in combination with *Bv*4687. Thus, *masH* was replaced by *Bv*4687 in further co-expression assay. The artificial gene clusters were then subcloned to the destination vector and resulted in three modules for better flexibility of adjusting composition, including pET-22a(+)-*masD-masE-masF-masG*, pACYCduet-*Bv*4687 (*masH* homolog from *B. vietnamiensis* LMG 10929, Genbank accession number WP_011881350)-*masI-masJ*, and pRSFduet-*masB*. Two or three modules were co-transformed to *E. coli* strain C41 (Yeastern Biotech, Taiwan), and selected colonies were then inoculated in LB medium with suitable antibiotics. The expression was induced by adding isopropyl-β-D-thiogalactopyranoside (IPTG) and cultured for two days. The extractions were carried out by adding equal volumes of EtOAc and sonicated for 30 min. Finally, the extracts were concentrated and transferred to DMSO for further analysis.

**Minimum inhibitory concentration determination and genetic rescue assay.** The minimum inhibitory concentration (MIC) measurement was modified from R J Lambert's method[38]. Different concentrations of collimonin C **1**, collimonin D **2**, massilin A **3**, and amphotericin B were prepared in YPD. The *C. albicans* cell viabilities were seeded with initial $OD_{600}$ 0.05 and incubated at 37˚C. After 24 h incubation, the final $OD_{600}$ was recorded by Epoch 2 Microplate (BioTek Instruments, USA) for MIC calculation. The MIC of polyynes was built with cell viability (%) of different concentrations, fitting into the modified Gompertz function[38].

For the genetic rescue assay, the *ERG10* overexpression and *masL* heterologous expression strains (Supplementary Fig. 41) were seeded with $OD_{600}$ 0.05 in YPD treated with MIC of each polyyne at 37˚C and supplied with 40 μg/mL doxycycline for gene expression. The *C. albicans* cell viabilities were recorded at 24 h. The statistical results were analyzed using GraphPad Prism (Version 8, GraphPad Software, USA) with multiple *t*-test analyses (FDR < 0.05).

**Enzymatic inhibition assays.** The enzymatic inhibition assay of acetyl-CoA acetyltransferase was conducted using a modified method[39] (see Supplementary Methods for details). The releasing CoA was monitored using a fluorescent probe. The fluorescence intensity of each experiment was obtained by subtracting the fluorescence intensity of the polyyne-free reaction. The measurement and calculation of polyynes inhibition kinetic refer to the previous covalent inhibitor model[40]. The progress curves used the hyperbolic regression model to calculate the kinetic inhibition parameters using GraphPad Prism (Version 8, GraphPad Software, USA).

**Protein expression and crystallization.** The recombinant MasL protein was purified from *E. coli* cells (see Supplementary Methods for details). For MasL-collimonins complex, 20 μM MasL was incubated with 100 μM collimonin C **1** or D **2** for further purification and crystallization. A freshly thawed aliquot of MasL, MasL-collimonin C complex, and MasL-collimonin D complex was concentrated to 20 mg/mL for an initial crystallization screening of ca. 500 conditions (Academia Sinica Protein Clinic, Academia Sinica). X-ray diffraction experiments were conducted by the National Synchrotron Radiation Research Center (Hsinchu, Taiwan) at the TLS beamline 15 A or the TPS beamline 05 A. The MasL, MasL-collimonin C, and MasL-collimonin D complex structures were solved by the molecular replacement method using the structure of thiolase from *Clostridium acetobutylicum* (pdb ID **4N44**) as the search model. The molecular figures were produced with Maestro (Schrödinger Release 2021-1: Maestro, Schrödinger, USA).

**Statistics and reproducibility.** For Fig. 4, the cell viabilities were normalized to the mock treatment. The standard deviation was calculated based on three replicates and the two-tailed Student's *t*-test for statistical analysis. The experiments in this study, including *mas* co-expression in *E. coli*, minimum inhibitory

concentration determination, genetic rescue assay, and enzymatic inhibition assay, were performed in three biological replicates (*N* = 3).

**Reporting summary.** Further information on research design is available in the Nature Research Reporting Summary linked to this article.

## Data availability

The transcriptome datasets were deposited in the Sequence Read Archive (SRA) database under the accession number **PRJNA706894**. The refined models of MasL, MasL-collimonin C, and MasL-collimonin D complex were deposited in the Protein Data Bank with pdb codes **7EI3**, **7EI4**, and **7FEA**, respectively. Detailed experimental methods are described in the supplementary information. The results of transcriptome analysis, Multigeneblast, and bottom-up proteome were listed in Supplementary Data 1, 2, and 3.

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

## Acknowledgements

This research was funded by the Ministry of Science and Technology of Taiwan (MOST 104-2320-B-001-019-MY2). We thank Prof. Chih-Chuang Liaw (National Sun Yat-sen University, Taiwan) and the R/V Ocean Researcher III team for collecting marine sediment. We thank Dr. Chao-Jen Shih (Bioresource Collection and Research Center, Taiwan) for isolating and identifying *Massilia* sp. YMA4. The materials and methods for constructing biosynthetic gene-null mutant strains were generously provided by Prof. Nai-Chun Lin's Lab (National Taiwan University, Taiwan). We thank Prof. Ching-Hsuan Lin and Ms. Chih-Chieh Hsu (National Taiwan University, Taiwan) for providing the material for constructing a tetracycline-inducible expression system. We thank Mr. Ning Lu, Ms. Ying-Mi Lai, and Ms. Chia-Chi Peng for collecting preliminary data. NMR data were collected in the High Field Nuclear Magnetic Resonance Center, Academia Sinica. LC-MS data were collected in the Metabolomics Core Facility, Agricultural Biotechnology Research Center, Academia Sinica, and the Proteomics Core Laboratory, Institute of Plant and Microbial Biology, Academia Sinica. TEM data were collected in the Biological Electron Microscopy Core Facility, Academia Sinica. The EM core facility was funded by the Academia Sinica Core Facility and Innovative Instrument Project (AS-CFII-108–119). We further thank the Protein Crystallization Facility (Academia Sinica Protein Clinic, Academia Sinica) for crystallization preparation; and the National Synchrotron Radiation Research Center (Hsinchu, Taiwan) with beamlines TLS 15 A and TPS 05 A for assistance in X-ray data collection and access to the synchrotron radiation centers.

## Author contributions

C.-C. L., S. Y. H., L.-T. M., C. L., C.-H. S., H.-J. L., P.-Y. C., L.-J. S., B.-W. W., and W.-C. H. performed the experiments. C.-C. L., S. Y. H., K.-F. H., T.-P. K., and Y.-N. H. carried out the data analysis. C.-C. L., S. Y. H., and L.-T. M. wrote the manuscript. Y.-L. Y. supervised the study and acquired funding to support the work.

## Competing interests

There are no conflicts to declare.
