## [Peer Review File · Communications Biology]

Reviewers' comments:

Reviewer #1 (Remarks to the Author):

Lin et al. describe the report of the discovery, identification, biosynthesis, and mode-of-action of bacterial polyynes produced by *Massilia* sp. YMA4. The main text was logical, and many parts of the authors' argument were reasonably supported by the results obtained. I judge that this paper has the potential to become a major milestone in bacterial polyynone research. However, I'd like to require revisions in the following points, and if these are achieved, I'll consider the paper worthy of acceptance.

1. Although this is a biological journal, I think that the details of the NMR-based structure determination of massilins need to be added to the main text or support.
2. P5,L27. The sentence "Rubredoxin-hydroxylase fusion ... to polyynone production." is not well understood and makes it difficult for the reader to predict how rubredoxin is involved in polyynone biosynthesis.
3. P6,L6. In the heterologous expression system in *E. coli*, polyynones were obtained even in the absence of *masL*, but was there any effect on the growth of *E. coli* in that case? Please add more information in the revised version.
4. P6,L15. It is possible that *malL* may be involved in polyynone biosynthesis and metabolic degradation, but how can such a possibility be ruled out. Please provide additional information.
5. P7,L32. You have proposed the importance of hydroxy groups in the polyynones, but please validate your conclusions by discussing the relevance of the polyynones without hydroxy groups showing antifungal properties.

Reviewer #2 (Remarks to the Author):

Research into bacterial polyynone is an area that has received a recent uptick over the last few years, and this manuscript addresses an important question surrounding the self-resistance and target of massilin and collimonin polyynones that exhibit antimicrobial activity against fungi – an activity shared by multiple bacterial polyynones. The manuscript is highly engaging, and the investigatory procedure nicely leads the reader through the experiments and results presented. I am confident in the conclusions reached in this manuscript are appropriate. However, the manuscript lacks significant discussion and reference to the literature, which subtracts from the manuscript. The definition of core polyynone biosynthetic genes has been demonstrated previously in the literature, but the authors have not referenced this in the manuscript – while it does subtract from the originality of this particular result, inclusion of key references is important to corroborate existing findings in the literature. In contrast, the expression of the core genes in a heterologous host is a novel addition to the literature and builds nicely on existing research. Some insights into mechanisms of activity of polyynones (proteogencin) against algal cells was demonstrated by Hotter et al. 2021, but this was also not discussed.

Please find below general comments for the manuscript alongside section specific comments that should be addressed to improve and rebalance the manuscript.

General comments

The results and discussion section is weighted towards results with minimal discussion and reference to the literature. Given the interest in bacterial polyynones in recent years, with publications on bioactivity of newly characterised polyynones and polyynone evolutionary history and phylogenetics, the discussion should be expanded. Where relevant, I have suggested key publications that should be included and discussed although this is not an exhaustive list, and the authors should review and enhance their discussion.

Additional methods are provided in the supplementary, however, these are seldom referred to in the main methods section. Please add "see supplementary methods" where applicable to direct the reader to further details. Additionally, a leading sentence or two for the main methods section that describes which methods are solely available in the supplementary methods would be a useful addition to the manuscript.

The manuscript incorrectly uses the Δ symbol to represent the mutants generated. The Δ symbol indicates a gene deletion, e.g. YMA4 Δ masF indicates a deletion of the masF gene, while the manuscript actually refers to gene inactivation by insertional mutagenesis i.e. YMA4::masF. This also highlights the issue that insertional mutations can have polar effects on downstream genes, as such sequential insertional mutants as described in the manuscript and subsequent loss of polyynes biosynthesis may not indicate essentiality of a gene as downstream genes may also be impacted in their expression. Please clarify this in the manuscript.

Multiple supplementary figures, e.g. COSY, NMR etc figures are not referenced in the main text or supplementary material.

Introduction:

I would advise against the use of exact numbers when describing polyynes metabolites. Given the recent interest in polyynes research, and the variability in defining polyynes derivatives as distinct polyynes vs distinct polyynes BGCs, this number is subject to change. I would suggest rephrasing to "more than x polyynes have been recorded". The introduction is concise but could benefit from a more detailed background to bacterial polyynes: Inclusion of the antimicrobial activity observed for caryoynencin in the first paragraph which includes bioactivity of other polyynes. Mention of ergoyne polyynes from *Gyvuella sunshinyii* or fischerellins from *Fischerella muscicola*, although no characterised activity.

The reclassification of *Pseudomonas cepacia* to *Burkholderia diffusa* applies specifically to the strain from which cepacins were originally isolated: SC 11783. Please include the strain to make this distinction.

Results and discussion

Transcriptomics analysis reveals polyynes as antifungal agents and their encoding BGC in *Massilia* sp. YMA4

What is the reason for mentioning supplementary data 2 in the legend of supplementary Fig 1?

There is mention of the antifungal metabolites (polyynes) being "unstable in the extract and hard to purify". Later in this section, after the metabolites have been identified as polyynes, this would be an opportune moment to refer to the literature as several other publications have commented on the difficulty and skill required to purify polyynes metabolites e.g. Reference 14: Ross et al. 2014 regarding caryoynencin purification with click chemistry; Reference 6: Kai et al. 2018 regarding collimonins with exclusion of oxygen and light etc.; and Reference 8: Mullins et al. 2021 regarding the purification of protegencin (protegenin) without requiring click chemistry. This would help build a more substantial discussion.

This sentence is ambiguous: "Among the four polyynes, collimonin C 1 and collimonin D 2 were isolated from *C. fungivorans* Ter331". Please rephrase to something like: Among the four polyynes isolated from *Massilia* sp. YMA4, were the metabolites collimonin C 1 and collimonin D 2, originally isolated from *C. fungivorans* Ter3316.

Again, the mention of antifungal activity of massilin A and collimonins C and D could be complemented with discussion.

Phylogenetic analysis of polyynes BGCs and mas heterologous co-expression revealed the core components for polyynes biosynthesis

A key reference is missing in the results/discussion: Defining the core biosynthetic module of polyynes has been published previously in the literature, however, this is not mentioned in the results and discussion section. Please include this reference (Mullins et al. 2021. Discovery of the *Pseudomonas* Polyynes Protegencin by a Phylogeny-Guided Study of Polyynes Biosynthetic Gene Cluster Diversity), which has been previously used in the manuscript, as this will also corroborate the results shown in this manuscript.

Manuscript refers to insertional mutagenesis but uses the delta symbol that actually refers to clean

deletions. Important distinction as insertional mutations could cause polar effects. See general comments. Additionally, Mullins et al. 2020 shows the essentiality of desaturase proteins and thioesterase in the formation of polyynes, but is not discussed in relation to the results presented here.

Great concept to express core genes in a heterologous host system and detect basic polyynes structure.

“In order to verify the in-silico result, we use plasmid-derived insertion mutagenesis to gain a serial mutant of mas BGC from masD (FAAL) to masL (Supplementary Figure 2)”. However, this figure does not include masG. Please indicate that masG (encoding an acyl carrier protein) was not included in this.

Check italicisation of genes and biosynthetic gene cluster names e.g. mas in manuscript

MasL serves as a polyynes direct target with a protective function
Interesting observation of the two chemotaxonomy groups of polyynes by phylogenetics!

Gene clusters are mentioned but not referenced e.g. ccn BGC – also please include the full name when mentioning the BGCs the first time to provide context e.g. cepacin (ccn).

The ccn BGC as ancestral to mas/col BGCs is also corroborated by Mullins et al. 2021.

Impressive protection of *C. albicans* by expression masL self-resistance gene – really nice demonstration of resistance gene.

SRG abbreviation only used twice – consider removing?

Candida albicans. ERG10 is the antifungal target of polyynes

The authors show that the masL homologue in *C. albicans*, ERG10, is the target of collimonin/massilin, and that ERG10 is involved in ergosterol biosynthesis – crucial to fungal cell membrane formation. The TEM images (Supplementary Figure 19) show an impact on the cell membrane following exposure to polyynes. However, is the cell membrane disruption a direct action of polyynes as suggested in this section and the abstract, or a consequence of ERG10 inhibition? The rescue of *C. albicans* viability through overexpression of the ERG10 gene suggests that polyynes are not directly disrupting the cell membrane which supposedly would also reduce cell viability if a direct target of polyynes. Can the authors clarify this statement in this section and the abstract. In addition, Hotter et al. 2021 investigated the mechanism of algicidal activity of protegencin (protegenin), and cell membrane integrity which may help clarify/discuss the finding in this manuscript.

Methods

Bioinformatics analysis

How were the reads mapped to the *Massilia* sp. YMA4 genome?

Construction of mas co-expression platform in *Escherichia coli*

Why was the masH homologue from *B. vietnamiensis* used instead of the massilin BGC masH gene? This may have been explained elsewhere in the manuscript, but if not, could this be included.

Supplementary Methods

UPLC-DAD-MS/MS methods

Please add how many cultured agar plates were used for the extraction of massilins.

Reviewer #3 (Remarks to the Author):

The manuscript by Lin et al describes discovery of a new polyynes metabolite and detailed

characterization thereof including the identification of its biosynthetic gene cluster (BGC), self-resistance mechanism, as well as the molecular origin of its antifungal activity.

Their work was initiated with the observation that *Massilia* sp. YMA4 appears to produce antifungal metabolites against *C. albicans*. Since a classical bioactivity-guided isolation approach was not successful, presumably due to the instability of the metabolite, they undertook an omics approach based on an *in silico* prediction of BGCs followed by gene regulation profiling. This strategy worked beautifully to reveal the *mas* gene cluster, which provided crucial information with the authors allowing them to successfully isolate four polyynone metabolites, collimonin C/D and massilin A/B. Sequence analysis as well as heterologous expression of *mas* genes confirmed that these metabolites indeed originated from the *mas* gene cluster. Furthermore, phylogenetic analysis led them to hypothesize that MasL, a homolog of acetyl-CoA acetyltransferase, is likely to be responsible for self-resistance. This idea could be unequivocally verified by obtaining co-crystal structure of MasL and collimonin C/D, in which a covalent linkage was formed between an active site Cys and the polyynone moiety. This observation suggested that the isolated polyynone metabolites could function as covalent acetyl-CoA acetyltransferase inhibitors. In fact, the authors found that collimonin C/D were capable of forming an analogous covalent adduct with ERG10, an acetyl-CoA acetyltransferase homolog in *C. albicans*, which not only confirmed their hypothesis on the mode of action, but also suggested that this mechanism would be responsible for their antifungal activity.

Overall, in my opinion, this is a beautiful piece of work that has effectively amalgamated many different scientific disciplines including omics analysis, natural product chemistry, structural crystallography, and microbiology in unveiling the structural identity as well as functional mechanism of new polyynone metabolites. All data were concisely well presented in the main text, and all experimental details were effectively presented in comprehensive SI. As indicated in the introduction, polyynone metabolites are structurally intriguing class of natural products. Not only this entity is very rare, but also there is only limited information on their mode of biological action. In this regard, the finding that the polyynone moiety can function as an warhead to form a covalent linkage with an active site cysteine of acetyl-CoA acetyltransferases is certainly intriguing, and it will certainly draw great attention from researchers interested in natural product chemistry as well as the identification and application of bioactive molecules. Other than some typo issues, I do not have any complaint about this manuscript. Thus, I am happy to recommend this manuscript to be published as it is.

#reviewer 1

1. Although this is a biological journal, I think that the details of the NMR-based structure determination of massilins need to be added to the main text or support.

Response:

Thank you for the comment. We have added the indication of NMR-based structure determination of massilins in the Results and Discussion section **“Transcriptomics analysis reveals polyynes as antifungal agents and their encoding BGC in *Massilia* sp. YMA4”**. (2nd paragraph, L12-15)

“Their structures were elucidated using high-resolution mass spectrometry (**Supplementary Figures 3 and 4**) and nuclear magnetic resonance (**Supplementary Tables 1-5 and Supplementary Figures 23-42**; the detailed isolation procedure and structure elucidation are described in **Supplementary Methods**).”

The detailed NMR-based structure elucidation is provided in the Supplementary Methods section under the subheading: **Isolation, structure elucidation, and quantification of polyynes in *Massilia* sp. YMA4**.

2. P5,L27. The sentence “Rubredoxin-hydroxylase fusion... to polyne production.” is not well understood and makes it difficult for the reader to predict how rubredoxin is involved in polyne biosynthesis.

Response:

Thank you for the comment. This sentence has been revised for better understanding in the Results and Discussion section **“Phylogenetic analysis of polyne BGCs and *mas* heterologous co-expression revealed the core components of polyne biosynthesis”**.

“As an electron transporter, rubredoxin can shuttle reducing molecules from NAD(P)H to membrane-bound alkane hydroxylases²², which is likely crucial to activation/regeneration of acetylenase/desaturase for the poly-dehydrogenation process in polyne biosynthesis.”

3. P6,L6. In the heterologous expression system in *E. coli*, polyynes were obtained even in the absence of *masL*, but was there any effect on the growth of *E. coli* in that case? Please add more information in the revised version.

Response:

In our preliminary assay for heterologous co-expression, *E. coli* was less sensitive to polyne. Polyynes directly inhibit acetyl-CoA acetyltransferase in the mevalonate biosynthesis (MVA) pathway to synthesize isopentenyl diphosphate (IPP), the precursor of ergosterol biosynthesis. *E. coli* is a gram-negative bacterium that uses an alternative biosynthesis pathway, the methyl erythritol phosphate (MEP) pathway, to synthesize IPP. This would explain why *E. coli* was less sensitive to heterologously biosynthetic polyne even when *masL* was absent.

The information above has been added to the Results and Discussion section **“MasL serves as a polyne direct target with a protective function”**. (3rd paragraph, L9-12)

“This targeting manner might explain why the growth of *mas* expressed *E. coli* is unaffected when producing polyynes without *masL* because *E. coli* has an acetyl-CoA acetyltransferase-independent route, the methylerythritol phosphate (also called deoxyxylulose 5-phosphate) pathway, to produce isoprenoid.”

4. P6,L15. It is possible that malL may be involved in polyene biosynthesis and metabolic degradation, but how can such a possibility be ruled out. Please provide additional information.

Response:

We think MasL may not be involved in polyene biosynthesis since there is no significant change in the polyene production of *masL* null mutant strain compared to the wild type (**Supplementary Figure 2**). Second, the acetyl-CoA-acetyltransferase gene was not conserved in polyene BGCs mining across bacteria (**Supplementary Figure 6**). Third, in addition to *masL*, five other copies of acetyl-CoA-acetyltransferase (thiolase) genes were annotated in the whole genome of *Massilia* sp. YMA4, which may be involved in the biosynthesis of fatty acid. Therefore, the information above suggests that MasL is not crucial in the biosynthesis process of polyenes. Thus, we further corroborated and defined MasL as an additional protein for resistance against polyene self-toxicity in this study. However, we agree that MasL is potentially involved in metabolic degradation. The irreversible covalent binding of the polyene-MasL complex would be one type of degradation. MasL reduced the accumulation of polyenes even though it is an energy-exhausting strategy.

5. P7,L32. You have proposed the importance of hydroxy groups in the polyenes, but please validate your conclusions by discussing the relevance of the polyenes without hydroxy groups showing antifungal properties.

Response:

Thank you for the suggestion. We have revised the Results and Discussion **“Polyenes are covalent inhibitors of acetyl-CoA acetyltransferase”**. (3rd paragraph, L9-19)

“In the further comparative analysis of the polar interactions in the activation pocket of MasL in MasL-collimonin C and MasL-collimonin D complex, the C(7)OH moiety of collimonin C/D **1, 2**, His158 of MasL, and a water molecule formed a strong polar interaction network, including a direct hydrogen bond (3.00-3.16 Å) and a water-mediated hydrogen bond between C(7)OH and His158 (**Figure 5**). Furthermore, although there was no significant induced-fit within the pocket, the collimonin C/D **1, 2** caused the Arg135 on the tetramerization loop to swap and form a hydrogen bond interaction and salt bridge, respectively, across the two subunits within the binding site (**Figure 5**). These polar interacting residues for the collimonin C/D **1, 2** (inhibitor) binding were similar to CoA (product) in other thiolase models³¹. Therefore, the acetyl-CoA acetyltransferase inhibitions by collimonin C/D **1, 2** are competitive binding models against substrate acetyl-CoA”

We have further provided the massilin C **5** (polyene without hydroxyl group) inhibition kinetic and *in vivo* cell viability assay results in the revised **Figure 4** and **Table 1**, which indicate that massilin C **5** would possess different antifungal targets. The discussion has been added to the Results and Discussion section **“Candida albicans ERG10 is the antifungal target of polyenes”**. Data are shown in **Supplementary Figures 5** and **16**.

#reviewer 2

General comments

1. However, the manuscript lacks significant discussion and reference to the literature, which subtracts from the manuscript. The definition of core polyene biosynthetic genes has been demonstrated previously in the literature, but the authors have not referenced this in the manuscript – while it does subtract from the originality of this particular result, inclusion of key references is important to corroborate existing findings in the literature. In contrast, the expression of the core genes in a heterologous host is a novel addition to the literature and builds nicely on existing research. Some insights into mechanisms of activity of polyenes (protegenin) against algal cells was demonstrated by Hotter et al. 2021, but this was also not discussed.

Response:

We appreciate the originality of existing findings in the literature and did not mean to subtract any from them. The related references and discussion have been added to the article.

The references defining the core polyene BGC have been added to the Results and Discussion section **“Phylogenetic analysis of polyene BGCs and mass heterologous co-expression revealed the core components for polyene biosynthesis”**. (1st paragraph)

“The *in-silico* mining and comparative studies of polyenes/ terminal alkyne containing BGCs indicated that *jam*ABC homologs (fatty acyl-AMP ligase (FAAL), fatty acid desaturase (FAD), and acyl carrier protein (ACP)) in jamaicamides (*jam*) BGC would be the minimal module for polyenes biosynthesis^{18,19}. In addition, the gene cluster K in *C. fungivorans* Ter331 (collimonin BGC, which is abbreviated to col BGC) recruited not only FAAL, FAD, and ACP but also hydrolase/thioesterase (H/TE) and rubredoxin (Rd)^{6,7}. Therefore, we further explored the architecture of 56 bacterial polyenes BGCs with mas BGC as a reference to confirm the conserved composition of polyene biosynthetic associated enzymes. It contains the FAAL – 2x FAD – ACP – FAD – H/TE – Rd (**Supplementary Figure 6**). The essential proteins, including FAAL, FAD, and H/TE in the conserved region of polyene BGCs, have been confirmed in the biosynthesis of caryophycin in *B. caryophylli*¹⁶, cepacins in *Burkholderia ambifaria*⁴, and protegenins in *P. protegens*^{12,13}.”

The activity of protegenin against algal cells and relevant discussion are also included in the Results and Discussion section **“Candida albicans ERG10 is the antifungal target of polyene”**. (L11-20)

“In the TEM analysis, we also observed that bacterial polyenes disrupted the cell membrane integrity in *C. albicans* (**Supplementary Figure 18**). This result corresponded to the earlier finding that protegenin (or protegenin A) permeated algal cells and caused cell lysis¹⁴. Notably, massilin C 5, the analog of protegenin (or protegenin A), possessed antifungal activity against *C. albicans* (MIC 12.22 μ M, **Supplementary Figure 5**) and a comparable ERG10 inhibitory potency to polyenes 1-3 (**Supplementary Figure 16**). However, neither the heterologous expression of MasL nor ERG10 in *C. albicans* was able to recover the cell viability under massilin C 5 treatment (**Figure 4**). Together these results suggested that the polyenes without a hydroxy moiety would possess multiple antifungal targets”

2. Additional methods are provided in the supplementary, however, these are seldom referred to in the main methods section. Please add “see supplementary methods” where applicable to direct the reader to further details. Additionally, a leading sentence or two for the main methods section that describes which methods are solely available in the supplementary methods would be a useful addition to the manuscript.

Response:

Thank you for the comment, “see Supplementary Methods” has been added to the manuscript where relevant.

3. The manuscript incorrectly uses the Δ symbol to represent the mutants generated. The Δ symbol indicates a gene deletion, e.g. YMA4 Δ masF indicates a deletion of the masF gene, while the manuscript actually refers to gene inactivation by insertional mutagenesis i.e. YMA4::masF. This also highlights the issue that insertional mutations can have polar effects on downstream genes, as such sequential insertional mutants as described in the manuscript and subsequent loss of polyene biosynthesis may not indicate essentiality of a gene as downstream genes may also be impacted in their expression. Please clarify this in the manuscript.

Response:

Thank you for your kind comment. We have corrected the symbol of the mutant throughout, as suggested.

As you mentioned, we indeed found that insertional mutation might not be suitable for well-defined core genes of polyene BGCs due to the polar effect (<https://doi.org/10.1371/journal.pone.0132657>), which is also supported by our qPCR results. The insertion mutation of the target gene in mas BGC would disrupt the expression of the downstream gene. That is why we further heterologously coexpressed core genes of mas BGC in *E. coli* to complement the mutagenesis results.

4. Multiple supplementary figures, e.g., COSY, NMR, etc. figures are not referenced in the main text or supplementary material.

Response:

Thank you for the comment. These figures have been referenced in the Results and Discussion section **“Transcriptomics analysis reveals polyenes as antifungal agents and their encoding BGC in *Massilia* sp. YMA4”. (2nd paragraph, L12-15)**

“Their structures were elucidated using high-resolution mass spectrometry (**Supplementary Figures 3 and 4**) and nuclear magnetic resonance (**Supplementary Tables 1-5 and Supplementary Figures 23-42**; the detailed isolation procedure and structure elucidation are described in **Supplementary Methods**)”

5. I would advise against the use of exact numbers when describing polyene metabolites. Given the recent interest in polyene research and the variability in defining polyene derivatives as distinct polyenes vs distinct polyene BGCs, this number is subject to change. I would suggest rephrasing to “more than x polyenes have been recorded”. The introduction is concise but could benefit from a more detailed background to bacterial polyenes: Inclusion of the antimicrobial activity observed for caryophycin in the first paragraph which includes bioactivity of other polyenes. Mention of ergoyne polyenes from *Gynerella sunshinyi* or fischerellins from *Fischerella muscicola*, although no characterised activity.

Response:

Thank you for the suggestion. We agree that the number of discovered polyynes is subject to change, so the sentence has been rephrased. The rephrased sentence in Introduction section is as follows:

“This instability has discouraged surveys of bacterial polyynes using a bioactivity-guided isolation approach. Nevertheless, more than ten bacterial polyynes have been recorded in a few species. Notably, most polyynes have been reported to have a broad spectrum of antimicrobial effects.” (L6-9)

We are also grateful for the comment about the lost references for bioactivity of caryoynencin and other polyynes and for the advice to add more reported polyyne research. We have included that information and references in the Introduction section as follows:

“Caryoynencin, isolated from *Pseudomonas caryophylli* (taxonomically reclassified as *Burkholderia caryophylli*), was reported to have antibacterial activity against *Escherichia coli*, *Klebsiella pneumoniae*, *Staphylococcus aureus*, and *Bacillus subtilis*.” (L12-15)

“Other polyynes, such as Ergoynes isolated *Gynuella sunshinyii* and fischerellins from *Fischerella muscicola* have potential cytotoxic/antibacterial and allelopathic abilities.” (L18-20)

6. The reclassification of *pseudomonas cepacia* to *Burkholderia diffusa* applies specifically to the strain from which cepacins were originally isolated: SC 11783. Please include the strain to make this distinction.

Response:

Thank you for the reminder to add the strain name to describe the polyyne-produced bacteria better. We have included the strain name in the Introduction section as follows:

“For instance, cepacins, isolated from *Pseudomonas cepacia* SC 11783 (taxonomically reclassified as a *Burkholderia diffusa*), was reported to have antibacterial activity against the majority Gram-negative bacteria....” (first paragraph, L9-11)

7. Transcriptomics analysis reveals polyynes as antifungal agents and their encoding BGC in *Massilia* sp. YMA4 What is the reason for mentioning supplementary data 2 in the legend of Supplementary Fig 1?

Response:

We apologize for the mistake and have corrected it.

8. The antifungal metabolites (polyynes) are “unstable in the extract and hard to purify”. Later in this section, after the metabolites have been identified as polyynes, this would be an opportune moment to refer to the literature as several other publications have commented on the difficulty and skill required to purify polyyne metabolites, e.g., Reference 14: Ross et al. 2014 regarding caryoynencin purification with click chemistry; Reference 6: Kai et al. 2018 regarding collimonins with exclusion of oxygen and light, etc.; and Reference 8: Mullins et al. 2021 regarding the purification of protegenin (protegenin) without requiring click chemistry. This would help build a more substantial discussion.

Response:

Thank you for the suggestion. We have modified the sentence and added a description and corresponding references concerning the difficulty and skill necessary to purify and identify polyynes metabolites to the Results and Discussion **“Transcriptomics analysis reveals polyynes as antifungal agents and their encoding BGC in *Massilia* sp. YMA4” (L3-5)**

“In addition, the antifungal metabolites were unstable in the extract and laborious to purify for bioassays using the classic bioactivity-guided isolation approach, which includes purification with click chemistry, exclusion of oxygen and light, or careful optimization.”

9. **This sentence is ambiguous: “Among the four polyynes, collimonin C 1 and collimonin D 2 were isolated from *C. fungivorans* Ter331”. Please rephrase to something like: Among the four polyynes isolated from *Massilia* sp. YMA4, were the metabolites collimonin C 1 and collimonin D 2, originally isolated from *C. fungivorans* Ter3316.**

Response:

Thank you for the suggestion. This awkward sentence has been rephrased to “Among the four polyynes identified from *Massilia* sp. YMA4, collimonin C 1 and collimonin D 2 were initially isolated from *C. fungivorans* Ter331” (Results and Discussion section, 2nd paragraph, L15-17).

10. **Phylogenetic analysis of polyynes BGCs and mas heterologous co-expression revealed the core components for polyynes biosynthesis. A key reference is missing in the results/discussion: Defining the core biosynthetic module of polyynes has been published previously in the literature, however, this is not mentioned in the results and discussion section. Please include this reference (Mullins et al. 2021. Discovery of the *Pseudomonas* Polyynes Protegencin by a Phylogeny-Guided Study of Polyynes Biosynthetic Gene Cluster Diversity), which has been previously used in the manuscript, as this will also corroborate the results shown in this manuscript.**

Response:

We appreciate the comment, and the suggested references have been included in the Results and Discussion section **“Transcriptomics analysis reveals polyynes as antifungal agents and their encoding BGC in *Massilia* sp. YMA4”. (2nd paragraph, L5-8)**

“Furthermore, a study of polyynes BGC (*pgn* BGC) in *P. protegens* had determined that the mutation of homolog (*pgnH*) of *masH* and *masI* in *pgn* BGC can effectively disrupt the production of protegencin (also known as protegenin A).”

And also, in the Results and Discussion section **“Phylogenetic analysis of polyynes BGCs and mas heterologous co-expression revealed the core components for polyynes biosynthesis”. (2nd paragraph, L1-6)**

“The in-silico mining and comparative studies of polyynes/ terminal alkyne containing BGCs indicated that *jamABC* homologs (fatty acyl-AMP ligase (FAAL), fatty acid desaturase (FAD), and acyl carrier protein (ACP)) in jamaicamides (*jam*) BGC would be the minimal module for polyynes biosynthesis^{18,19}. In addition, the gene cluster K in *C. fungivorans* Ter331 (collimonin BGC, which is abbreviated to col BGC) recruited not only FAAL, FAD, and ACP but also hydrolase/thioesterase (H/TE) and rubredoxin (Rd).”

11. **Manuscript refers to insertional mutagenesis but uses the delta symbol that actually refers to clean deletions. Important distinction as insertional mutations could cause polar effects. See general comments. Additionally, Mullins et al. 2020 shoes the essentiality of desaturase proteins and thioesterase in the formation of polyynes, but is not discussed in relation to the results presented here.**

Response:

Thank you again for the suggestion. The related discussion of homologs of *masH* (desaturase) from *pgn* BGC has been added in the Results and Discussion section as follows:

“Furthermore, a study of polyynone BGC (*pgn* BGC) in *P. protegens* had determined that the mutation of homolog (*pgnH*) of *masH* and *masI* in *pgn* BGC can effectively disrupt the production of protegencin (also known as protegenin A).” (2nd paragraph, L5-8)

12. **“In order to verify the in-silico result, we use plasmid-derived insertion mutagenesis to gain a serial mutant of *mas* BGC from *masD* (FAAL) to *masL* (Supplementary Figure 2)”. However, this figure does not include *masG*. Please indicate that *masG* (encoding an acyl carrier protein) was not included in this. Check italicisation of genes and biosynthetic gene cluster names e.g. *mas* in manuscript**

Response:

Thank you for your suggestion. In this study, we did not construct the mutation of *masG* because ACPs (acyl carrier proteins) are generally reported as essential proteins for microbial fatty acid metabolism (DOI: 10.1186/s13068-016-0430-4) and the production of polyketides (DOI: 10.1016/j.isci.2020.100938). As bacterial polyynes mostly come with a long-chain fatty acid backbone, we assume that the precursor of bacterial polyynes may come from fatty acid metabolism and that *masG* is crucial for producing massilins and collimonins.

Moreover, we found that the insertion mutation of *mas* BGC comes with polar effects, as you mentioned. Thus, we further constructed a heterologous co-expression system in *E. coli*. The core genes of *mas* BGC were defined by heterologous co-expression in our study, indicating that *masG* is a core gene to produce polyynes.

The italicization of genes and biosynthetic gene cluster names has been double confirmed, and we appreciate the reminder.

13. **Gene clusters are mentioned but not referenced e.g. *ccn* BGC – also please include the full name when mentioning the BGCs the first time to provide context e.g. cepacin (*ccn*). The *ccn* BGC as ancestral to *mas/col* BGCs is also corroborated by Mullins et al. 2021.**

Response:

Thank you for the comment. We have added the full name of *ccn/col* when mentioning the BGCs for the first time.

14. **SRG abbreviation only used twice – consider removing?**

Response:

We have removed the SRG and marked it directly as a self-resistance gene in the Results and Discussion section as follows:

“MasL serves as a polyene direct target with a protective function”. (2nd paragraph, final sentence, and 3rd paragraph, L7)

15. *Candida albicans* ERG10 is the antifungal target of polyene. The authors show that the masL homolog in *C. albicans*, ERG10, is the target of collimonin/massilin, and that ERG10 is involved in ergosterol biosynthesis – crucial to fungal cell membrane formation. The TEM images (Supplementary Figure 19) show an impact on the cell membrane following exposure to polyenes. However, is the cell membrane disruption a direct action of polyenes as suggested in this section and the abstract, or a consequence of ERG10 inhibition? The rescue of *C. albicans* viability through overexpression of the ERG10 gene suggests that polyenes are not directly disrupting the cell membrane which supposedly would also reduce cell viability if a direct target of polyenes. Can the authors clarify this statement in this section and the abstract. In addition, Hotter et al. 2021 investigated the mechanism of algicidal activity of protegencin (protegenin), and cell membrane integrity which may help clarify/discuss the finding in this manuscript.

Response:

Thank you for your kind suggestion.

The TEM result of polyenes is similar to fluconazole (an antifungal drug), which targets ERG11 of the ergosterol biosynthesis pathway (doi:10.1128/AAC.00499-13). The ergosterol biosynthesis pathway is crucial for producing ergosterol to maintain the integrity of the cell membrane. Thus, we assumed that the bacterial polyenes inhibited ERG10 of the ergosterol biosynthesis pathway and consequently disrupted the cell membrane integrity and inhibited the cell viability of *C. albicans*. We have revised the sentences in the Results and Discussion section as follows: “*Candida albicans* ERG10 is the antifungal target of polyenes”) and also in the Abstract to clarify our findings.

On the other hand, we found a new compound, massilin C 5, an analog of protegencin (or protegenin A) with a shorter fatty acid chain. The antifungal effect of massilin C 5 against *C. albicans* cannot be recovered by overexpressing ERG10, which demonstrated that the hydroxyl group of polyenes is crucial for antifungal effects with target specificity on ERG10. Thus, we assume that polyenes without hydroxyl groups, such as massilin C 5 or protegencin (or protegenin A), might possess multiple antifungal mechanisms to disrupt the cell membrane integrity.

16. **Bioinformatics analysis. How were the reads mapped to the *Massilia* sp. YMA4 genome?**

Response:

Thank you for the comment. We have included detailed information in the Methods, “**Bioinformatics analysis**” section as follows:

“The RNA sequencing and transcriptome analysis were performed using Illumina MiSeq system (Illumina, USA) and CLC genomics workbench (version 11, CLC bio, Denmark), and the 92.89-98.11% reads were successfully mapped to the *Massilia* sp. YMA4 genome (GenBank assembly accession: GCA_003293715.1) for RPKM quantification (see **Supplementary Methods** and **Supplementary Data 1** for details).” (2nd paragraph, Lines 1-5)

17. **Construction of mas co-expression platform in *Escherichia coli*. Why was the *masH* homologue from *B. vietnamiensis* used instead of the massilin BGC *masH* gene? This may have been explained elsewhere in the manuscript, but if not, could this be included.**

Response:

Thank you for the comment. Detailed information is included in the Supplementary Methods. However, for better understanding, we have moved the detailed description to the main text in the Methods section **“Construction of *mas* co-expression platform in *Escherichia coli*”**. (1st paragraph)

“Individual *mas*and ligation. To obtain functional *masH* in the heterologous *E. coli* system, we recruited native and codon-optimized *masH* and the homolog (*Bv4687*) from the palmitate-derived (C16) species *B. vietnamiensis* LMG 10929 (see **Supplementary Figures 8 and 21**) into modular co-expression. The backbones of the constructed modules, including pET-22a(+), pACYCduet, and pRSFduet, were purchased from Merck (Germany). However, polyne accumulations were observed only in combination with *Bv4687*. Thus, *masH* was replaced by *Bv4687* in a further co-expression assay. The artificial gene clusters were the....”

18. **UPLC-DAD-MS/MS methods. Please add how many cultured agar plates were used for the extraction of massilins.**

Response:

Thank you for the comment. A detailed description has been added to the Supplementary Methods, “UPLC-DAD-MS/MS methods” section as follows:

“Approximately 5000 PDA agar plates were cultured with *Massilia* sp. YMA4 and extracted with EA 2-3 times. The extracts were concentrated and replaced with DMSO, preventing polyne degradation while drying or resuspending for LC-MS/MS analysis.” **(L1-3)**

#reviewer 3

We appreciate the reviewer’s comments and revised the manuscript thoroughly.

REVIEWERS' COMMENTS:

Reviewer #1 (Remarks to the Author):

I was able to judge that the authors were sincere in their responses to my revision request. The quality of the papers has been further improved with this revision, and I look forward to seeing the printed version.

Reviewer #2 (Remarks to the Author):

Having reviewed the rebuttal, I am pleased to see all comments and queries by reviewers have been addressed by the authors, and the manuscript updated accordingly.

I am happy with the author responses and recommend this manuscript for publication